# Silk Fibroin Coated Magnesium Oxide Nanospheres: A Biocompatible and Biodegradable Tool for Noninvasive Bioimaging Applications

**DOI:** 10.3390/nano11030695

**Published:** 2021-03-10

**Authors:** Jitao Li, Asma Khalid, Rajni Verma, Amanda Abraham, Farah Qazi, Xiuxiu Dong, Gaofeng Liang, Snjezana Tomljenovic-Hanic

**Affiliations:** 1School of Physics and Telecommunications Engineering, Zhoukou Normal University, Zhoukou 466001, China; lijitao@zknu.edu.cn; 2School of Physics, The University of Melbourne, Parkville 3010, Australia; rajni.verma@unimelb.edu.au (R.V.); farah.qazi@student.unimelb.edu.au (F.Q.); snjezana.thanic@unimelb.edu.au (S.T.-H.); 3ARC Centre of Excellence for Nanoscale BioPhotonics, School of Science, RMIT University, Melbourne, VIC 3001, Australia; Amanda.Abraham@rmit.edu.au; 4State Key Laboratory of Bioelectronics, School of Biological Science and Medical Engineering, Southeast University, Nanjing 210096, China; dongxx@ujs.edu.cn; 5Medical College, Henan University of Science and Technology, Luoyang 471023, China; lgfeng990448@haust.edu.cn

**Keywords:** silk fibroin, magnesium oxide, bioimaging, fluorescent nanoparticles

## Abstract

Fluorescent nanoparticles (NPs) have been increasingly studied as contrast agents for better understanding of biological processes at the cellular and molecular level. However, their use as bioimaging tools is strongly dependent on their optical emission as well as their biocompatibility. This work reports the fabrication and characterization of silk fibroin (SF) coated magnesium oxide (MgO) nanospheres, containing oxygen, Cr^3+^ and V^2+^ related optical defects, as a nontoxic and biodegradable hybrid platform for bioimaging applications. The MgO-SF spheres demonstrated enhanced emission efficiency compared to noncoated MgO NPs. Furthermore, SF sphere coating was found to overcome agglomeration limitations of the MgO NPs. The hybrid nanospheres were investigated as an in vitro bioimaging tool by recording their cellular uptake, trajectories, and mobility in human skin keratinocytes cells (HaCaT), human glioma cells (U87MG) and breast cancer cells (MCF7). Enhanced cellular uptake and improved intracellular mobilities of MgO-SF spheres compared to MgO NPs was demonstrated in three different cell lines. Validated infrared and bright emission of MgO-SF NP indicate their prospects for in vivo imaging. The results identify the potential of the hybrid MgO-SF nanospheres for bioimaging. This study may also open new avenues to optimize drug delivery through biodegradable silk and provide noninvasive functional imaging feedback on the therapeutic processes through fluorescent MgO.

## 1. Introduction

Fluorescent nanoparticles (NPs) are extensively investigated to monitor and track biological processes at a nanoscale level, via real-time bioimaging. The fluorescence from these NPs is anticipated to lead to advancements of diagnostic tools and targeted drug release therapies. Metal oxide NPs [1,2,3] have been gaining interest as contrast agent for several bioimaging applications owing to their room temperature single photon emission [1,3], tunable optical properties [4] and low toxicity. However, there are several limitations to the use of these metal oxide NPs, as some of them demonstrate low quantum efficiency and brightness [2,3], high agglomeration in cell culture media [2], while others exhibit dose dependent cytotoxicity [5]. One example is zinc oxide that exhibits broadband emission in the visible region, attributed to defects within the crystallographic structure, suitable for in vitro experiments. However, ZnO NPs generally suffer from pure photostability in cellular environment, in form of photobleaching or photo-blinking [1,3].

It is crucial for in vitro experiments for a fluorescent marker to absorb light at wavelengths longer than 500 nm and to emit light at wavelengths longer than 600 nm to avoid background autofluorescence of the cells [6]. For in vivo experiments, the light emission should be in near-infrared region (NIR), 700–900 nm, as it penetrates centimeters into tissue, whereas visible light can only travel microns [7].

Magnesium oxide (MgO) is a biologically since it is an essential mineral supplement used to prevent and treat low amounts of magnesium in the blood required for the normal functioning of cells, nerves, muscles, bones, and the heart [8]. In NP-form, MgO has extensive applications in biology including a catalytic agent for medicine, contrast agent for MRI imaging [9] and is widely studied for its antibacterial [10] and anticancer properties [11]. These NPs have a unique combination of biocompatibility and biodegradability in addition to being intrinsically fluorescent [2]. In a recent work [2], we have reported intrinsic photostable fluorescence properties of MgO NPs, from naturally occurring chromium Cr^3+^ and vanadium V^2+^ ions, in the visible to near infrared (NIR) range, used for real-time live cell tracking in cells derived from normal and cancerous tissues.

However, the main obstacle for future higher-level biomedical applications of MgO NPs are agglomeration and need for more effective cell internalization. Currently there is no fluorescent nanoparticle that satisfies all necessary criteria for short-term in vivo imaging: biocompatibility, biodegradability, photostability, suitable wavelengths of absorbance, fluorescence that differ from tissue autofluorescence, high internalization, and low level of agglomeration. In this study we test fluorescent MgO-silk fibroin (SF) nanoparticles that satisfy all the criteria.

In order to enhance the structural and optical properties of fluorescent NPs [12] as well as cellular uptake [13], polymer coatings have been used in several studies [12,13,14,15,16]. Moreover, polymeric particles and shells on their own have been investigated as targeted drug delivery vehicles to overcome the blood–brain barrier and deliver the drug more efficiently [16]. Both natural and synthetic materials can be used for NP coating as well as drug delivery vehicles. The limitation with synthetic polymers is that they are non-biodegradable and hydrophobic. They additionally require harsh acidic or organic solvents, which can induce cytotoxicity [16]. Naturally derived degradable polymers, such as collagen, gelatin, cellulose, hyaluronic acid, alginate, and chitosan have good biocompatibility but have uncontrollable degradation rates, show background autofluorescence and are either opaque or partially transparent to visible light [16]. Hence low transmission will block the NP-fluorescence and the polymer’s autofluorescence will introduce significant background noise during imaging.

SF proteins derived from *Bombyx mori* is an excellent biomaterial for bioimaging and drug delivery applications [14,15,16,17,18,19,20]. Regenerated SF solution extracted from the silkworm cocoons through an all-aqueous processing technique [20] demonstrates biocompatibility, excellent transparency (>85%) to visible and infrared light [14] and relatively high refractive index of 1.5414 [14]. SF degrades to simple amino acids which are absorbed by the body without causing an inflammatory reaction [14]. The SF protein can be transformed to a variety of structures and coatings [14,15,19,21,22] with tunable biodegradability [18]. In our earlier work we have reported NPs coated with optically transparent SF films, spheres and fibers for noninvasive, light based bioimaging [14,15], biosensing [22] and drug release [19] applications.

For the current manuscript, we fabricated and investigated a hybrid optical platform consisting of SF embedded MgO spheres. The surface and optical properties as well as the cellular uptake of the hybrid MgO-SF spheres were compared to bare MgO NPs. We report enhanced optical emission properties, room and body temperature fluorescence, reduced agglomeration, more effective cell internalization and cell imaging capability of MgO-SF spheres as compared to MgO NPs on their own. Furthermore, silk spheres degraded and showed a reduction in diameter when incubated at body temperature.

This opens the way for several future applications of hybrid dual biodegradable MgO-SF systems, since both MgO and SF are biodegradable. One of the potential applications of these novel biocompatible spheres is for bioresorbable traceable drug delivery agents. In such applications, it is envisioned that the fluorescent MgO will monitor and track the location of the spheres, while SF shells loaded with drugs could degrade in a sustained manner enabling controlled drug delivery.

## 2. Materials and Methods

### 2.1. Materials

Commercial MgO NP samples were purchased from MTI Corporation with an average size of 30 nm. The NP powder was dispersed into water and sonicated for 5 min to make a homogenous solution with a final concentration of 2 mg/mL. Polyvinyl alcohol (PVA) with a MW of 30,000–70,000 was obtained by Sigma Aldrich (St. Louis, MO, USA). Silk fibroin solution with a final concentration of 65 mg/mL was extracted and purified from *Bombyx Mori* silk cocoons as mentioned in the literature [20].

### 2.2. Synthesis of MgO-SF Spheres

For the fabrication of MgO-SF spheres a 2 mg/mL solution of MgO NPs was prepared. In total, 1 mL of SF aqueous solution (from 65 mg/mL as prepared concentration) was mixed with 5 mL of MgO aqueous solution (from 2 mg/mL concentration) and 0.5 mL of miliQ water (Merck KGaA, Darmstadt, Germany). The prepared mixture hence contained 10 mg of SF and ≈1.5 mg of MgO NPs, per mL of the solution, giving a weight percentage of 10:1 for SF versus MgO. MgO-SF spheres were fabricated through a home built co-flow microfluidic system as reported in our previous work [15]. The working solution of MgO-SF was injected through an inner smaller diameter (30G) needle of a co-flow device, dispensing at a discrete flow rate of 40 μL/h. The PVA with a 50 mg/mL concentration was loaded in a syringe with bigger diameter (16G) and dispensed at a continuous flow rate of 4 mL/h with programmable syringe pump (New Era Pump Systems, Inc., Farmingdale, NY, USA). The MgO-SF droplets condense to spheres upon contact with PVA stream and were collected in PVA for 10 h. The final concentration of collected spheres hence contained 400 µL of MgO-SF spheres, with ≈0.6 mg of MgO and 4 mg SF content. The spheres-suspension was dried overnight to allow the water to evaporate further decreasing sphere size. The PVA was completely removed via centrifugation twice in water at 40,000 RCF (g-force) for 20 min each at 4 °C. The washed spheres were then suspended and stored in 2–4 mL of water or phosphate buffered saline (PBS) (Sigma Aldrich, St. Louis, MO, USA).

#### Size-Reduced MgO-SF Spheres

The limitation of the microfluidic technique is that it produces spheres with size distribution of the order of 0.5–1 µm, which is not suitable for in vitro experiments. To reduce the size distribution, the spheres were suspended in PBS, followed by incubation and constantly shaking at 37 °C. This resulted in size-reduced spheres due to the degradation mechanism in SF. The degradation of the SF in PBS is attributed to bulk swelling of the SF causing silk molecules to detach from the surface reducing the diameter of the spheres over time [15,19]. The spheres were incubated for 48 h to make their size suitable for uptake by the cells.

### 2.3. Structural Characterization

Field-emission scanning electron microscopy (FESEM, Carl Zeiss Ultra Plus, Shanghai, China) was used to analyze the morphology and structure of hybrid spheres. The spheres and MgO NPs on their own in water were drop cast on silicon substrates. Each sample was sputter-coated with 5 nm of gold using an Electron Microscopy Sciences 300T dual head sputter coater. Sputtering was conducted at 20 mA with an average deposition rate of 4 nm/min.

### 2.4. Dynamic Light Scattering and Zeta Potential Measurements

The size and surface potential of the spheres were performed via a Malvern Zetasizer (ATA Scientific, Sydney, NSW, Australia). PBS was used as solvent to suspend the spheres and perform the measurements. The average was measured for six measurements repeated for each sample.

### 2.5. Confocal Imaging

A customized confocal microscope was used to explore the fluorescence properties of the MgO-SF and MgO NPs. Continuous wave (CW) lasers with excitation wavelengths of λ = 532 nm (green, Laser Quantum GEM 532 + SMD600 controller) and 633 nm (red, He-Ne) from Novanta Inc. (Fremont, CA, USA) were selected to separately excite the spheres and study their emission response. The laser beam was focused on the sample with a 100 × 0.9 NA objective lens. The same objective for excitation was used to collect the fluorescence. A combination of 532 nm laser line and 532 notch filters were used to separate the NP fluorescence and the excitation laser for green excitation mode. A 633 nm laser line and a 633 nm notch filter were used for the red excitation. The fluorescence was collected by a multimode fiber (GIF625, Thorlabs Inc., Newton, NJ, USA), and then delivered to avalanche photodiodes (SPCM-AQRH-14, Excelitas Technologies, Waltham, MA, USA) and a spectrometer (SpectraPro SP2500, Princeton Instruments Inc., Trenton, NJ, USA) for intensity and spectral measurement, respectively. The spatial resolution of the confocal microscope used is 0.3 µm. With the diffraction limited customized confocal microscope, nanoparticles with size range down to 10 nm can be scanned through their fluorescence.

Moreover, the confocal microscope we used for the experiment has a gaussian fitting function that maximizes the brightness of the single emitters/NPs with respect to their position. The gaussian fitting returns position co-ordinates x ≤ 0.3 µm, y x ≤ 0.3 µm for individual NPs. The x, y co-ordinates greater than 0.5 µm indicate aggregate of NPs or multiple fluorescent emitters inside the selected emitting spot. Hence the resolution of the microscope and the scanning software allow an indirect measurement for the dimensions of the spots as well as the number of NPs we analyzed within an emitting spot.

### 2.6. Cell Culture

Human skin keratinocytes (HaCaT), human glioma cells (U87MG) and human breast cancer cells (MCF7) were maintained in DMEM, high glucose culture medium, containing GlutaMAX and pyruvate (ThermoFisher Scientific, Melbourne, VIC, Australia), supplemented with 10% fetal bovine serum (ThermoFisher Scientific, VIC, Australia) and 1% penicillin-streptomycin (ThermoFisher Scientific, Melbourne, VIC, Australia). Cells were grown at 37 °C, in the presence of 5% CO_2_ and 95% relative humidity.

### 2.7. Cell Toxicity Analysis

Cell toxicity was assessed by resazurin assay. Cells were grown in 96-well plates for 24 h at a seeding density of 10,000 cells/well for the 24 h time point and 5000 cells/well for the 48 and 72 h time points. They were then treated with MgO NPs or MgO-SF nanospheres for 24–72 h, at concentrations ranging from 0 to 25 µg/mL. To assess viability, media was replaced with phenol red-free media, containing 15 µg/mL resazurin and cells were incubated for 3 h at 37 °C. The plates were then read on a fluorescence microplate reader at an excitation of 560 nm and emission at 590 nm. Statistical significance obtained by one-way ANOVA and Dunnett’s multiple comparison test were performed on GraphPad Prism 7.02 (© 2021 GraphPad Software).

### 2.8. Wide-Field Imaging

For tracking and imaging with the wide field, the NP-cultured cells were prepared. In total, 1 × 105 cells were seeded into glass bottom 35 mm dishes (Ibidi, Melbourne, VIC, Australia) and allowed to grow for 24 h. Cells were then treated with 5 µg/mL MgO NPs or MgO-SF spheres for 24–72 h. Cells were washed thrice in PBS and placed in phenol red-free DMEM complete media (ThermoFisher Scientific, Melbourne, VIC, Australia) before imaging at the respective time points. Cell nuclei were stained with NucBlueTM Live Ready Probes Reagent (ThermoFisher Scientific, Melbourne, VIC, Australia) for 20 min. Live cell imaging in real time was performed with a commercial wide-field fluorescence microscope (Olympus, Shinjuku City, Tokyo, Japan). The wide-field microscope has six LED sources, from which we used the 390 nm (violet) excitation to visualize the stained nuclei, 485 nm (blue) for SF’s fluorescence and 560 nm (green) to check the MgO and MgO-SF fluorescence inside the cells. We used the 40× magnification objective to scan the samples and the ultra-sensitive Hamamatsu Flash 4.0 Camera to record live imaging frames and movies.

To measure the diffusion coefficient, we recorded movies with the inbuilt camera of Olympus microscope. The movies were uploaded in MATLAB (R2017) computing software. Each movie comprised 200–300 frames taken for 60–90 s duration. A distribution of 8–12 particles in each frame was traced for the movie duration. The x_0_,y_0_ coordinates of the particles in the first frame were taken as origin. The displacement r_i_ = x_i_ − x_0_, … x_n_ − x_0_, of each particle in the successive frame was recorded with respect to its origin. Finally, the displacement squared ri^2^ was plotted as a function of time for each particle. The slope of the curve provided the diffusion coefficient or mobility of the particles inside the cells.

### 2.9. Staining Cells for Endosome Tracking

For endosome study, 5 × 104 cells/mL were seeded on glass coverslips in a 24-well plate and allowed to grow for about 8 h. Cells were transfected with CellLightTM Early Endosomes-GFP, BacMam 2.0 (ThermoFisher Scientific, Melbourne, VIC, Australia), at a concentration of 2 µL per 10,000 cells for 16 h at 37 °C. Cells were then treated with 5 µg/mL MgO NPs or MgO-SF nanospheres for 1, 2 and 4 h. Cells were stained with NucBlueTM Live Ready Probes Reagent for 20 min and then fixed in 4% (*v/v*) paraformaldehyde (ThermoFisher Scientific, Melbourne, VIC, Australia) in PBS for 10 min, at the various time points. The samples were mounted on to glass coverslips using ProLongTM Gold Antifade Mountant (TheroFisher Scientific, Melbourne, VIC, Australia) and imaged on an Olympus confocal microscope.

## 3. Results and Discussion

### 3.1. Structural Characteristics

FESEM measurements: The morphology and size of MgO and MgO-SF spheres were characterized by Field Emission Scanning Electron Microscopy (FESEM). Figure 1a,b represents FESEM micrographs of MgO NPs, where the bare NPs were observed to be highly agglomerated with a wide size distribution from 800 ± 60 nm to 2.40 ± 0.50 µm. Figure 1c,d shows the FESEM scans of MgO-SF spheres, clearly demonstrating that coating the MgO NPs with silk, yielded spherical particles with significantly reduced agglomeration and narrower size distribution from 1.54 ± 0.02 to 1.60 ± 0.02 µm. The particle size distribution of the MgO and as prepared hybrid MgO-SF spheres was calculated using MATLAB image sizing tool.

*DLS measurements:* The size distribution calculated via FESEM images was larger than that suitable for cell culture and imaging. Our previous work has reported the reduction in sphere size for hybrid silk particles upon incubation [13,17]. Hence the MgO-SF spheres produced on day 0 (shown in Figure 1) were incubated in PBS for 2 days (48 h) for size reduction. The DLS measurements were taken for the size reduced spheres before cell culture.

The size and zeta potential of the size-reduced MgO-SF spheres were determined to check the particle size distribution and stability against agglomeration under cell culture conditions, as shown in Table 1. A hydrodynamic size distribution less than 300 nm was observed for MgO-SF spheres which is suitable for cell culture [2], as shown in Table 1. The reduction in size upon incubation is owing to the degradation of silk spheres. The degradation is accelerated by the phosphate ions in PBS because the positive and negative charges more effectively interfere with the β-sheet structure of the silk molecules causing them to swell at different rates compared to water [15]. The DLS size distribution plot for MgO-SF spheres in PBS and 10% FBS + PBS spectra is shown in Appendix A. The SEM image for the size-reduced spheres is shown in Appendix A. These spheres showed a size distribution of 200 ± 50 nm.

Moreover, a negative zeta potential of −15.9 ± 0.8 mV was observed for MgO-SF spheres in PBS, which is expected to contribute to partial stabilization of these spheres in the solution and culture media by maintaining the electrostatic repulsion between the spheres [2]. In comparison MgO NPs on their own show significant agglomeration and a lower zeta potential of −11.4 ± 0.3 mV in PBS [2]. Moreover, silk fibrils consist of various positive and negative surface charges [18]. When NPs are added to SF solution, the silk fibrils connect electrostatically, and self-assemble around the NPs resulting in complete encapsulation of the NPs.

### 3.2. Spectral Properties

To determine the light absorption and emission properties, we performed the spectral analysis as shown in Figure 2. The characteristic absorption spectra were recorded for both MgO (blue) and MgO-SF (red) NPs, as demonstrated in Figure 2a. From the measured normalized absorbance of the particles in the visible to infrared range it can be seen that the absorbance increases at shorter wavelengths. However, both MgO NPs and MgO-SF spheres demonstrated absorption in the visible to infrared range. This implies that the NPs fulfil the requirements as probes capable of absorbing light at wavelengths longer than 500 nm are essential for low background bioimaging of cells and tissues. Moreover, the absorption trace (red, Figure 2a) for silk coated MgO spheres indicates that we can excite these hybrid NPs with absorption efficiency >60% (Figure 2a) in the entire wavelength range of 400–900 nm, which is highly crucial for in vivo experiments [7].

Emission properties were investigated using two excitation wavelengths, one at 532 nm (green laser) suitable for in vitro experiments and the second one at 633 nm (red laser) suitable for in vivo experiments. A comparison of photoluminescence (PL) was performed for MgO-SF spheres with MgO only NPs and SF only at excitation wavelength of 532 nm, as shown in Figure 2b. Although SF coated MgO spheres showed brighter florescence as compared to MgO NPs, the spectral features for the same color centers were identical. SF is highly optically transparent and the SF only spheres showed a weak autofluorescence as indicated by the black spectrum of Figure 2b.

The PL from MgO showed a broad spectrum in the visible to near infrared (NIR) range between 550 and 750 nm, with a maximum around 620 nm. The PL spectrum for MgO NPs were found to be similar when recorded at low temperature of −195 °C under 532 nm excitations (as shown in Appendix A). The common peak at 740 and 775 nm for the three samples in Figure 2b represent background PL from the glass (silica) substrate. These spectra from MgO NPs appeared to have different spectral characteristics than our previously reported data on MgO NPs [2].

Earlier literature reports that the MgO nanostructures with intrinsic defects such as oxygen/magnesium ion vacancies (F-center defects and V-center defects) mostly give a broad PL emission in the blue region at room temperature, while MgO with external defects (doped impurities), such as Cr, Fe, Mn or other transition metal ions doping, are responsible for red emissions [23,24].

A further detailed analysis of the MgO and MgO-SF samples under green (532 nm) excitation yielded additional emission centers with PL in the NIR region as shown in Figure 2c. For these less occurring fluorescent centers, we observed PL spectra shifted by 100 nm towards the NIR, compared to the Figure 2b spectra (under the same 532 nm excitation wavelength). These emission centers also demonstrated highly photostable counts (green data—Appendix A). The PL was observed in a 650–900 nm wavelength range, centered around 800 nm. Spectral features were observed at around 680, 700, 710, 725, 830, 850, and 860 nm. Hence, for the low occurring emission centers of Figure 2c,d, the spectral features below 800 nm are contributed by the Cr^3+^ ions. Whereas peaks visible on the right side of the broad band (above 800 nm) in Figure 2c are known to correspond to the V^2+^ ions. It has been shown that arc melting is known to introduce impurities of vanadium and chromium [2]. The inductively coupled plasma mass spectrometry (ICP-MS) on the MgO nanopowder confirms these findings [2].

When these less occurring centers were excited with red wavelength (633 nm), they showed a similar PL spectrum (compared to Figure 2c) but with enhanced peak intensities at 680, 700, 710, and 725 and a pronounced side band centered around 800 nm. Figure 2d represents emission spectrum of MgO-SF spheres excited at 633nm.

The section concludes that a multicolored fluorescence was observed in the MgO NP system. The multicolored experiments are often needed for molecular targeting or imaging of specific cells. With the MgO nanostructures we can perform multispectral imaging using combinations of different color centers, such as Cr^3+^, V^2+^ and O-vacancy related centers. Moreover, biosensors, which are capable of detecting multiple proteins and enzymes, have potential to accelerate diagnosis of diseases such as cancer. Current methods often only detect one protein per sample, requiring multiple tests to diagnose cancer and infectious disease.

From Figure 2b, approximately, 2× enhancement in fluorescence intensity was observed for SF coated MgO compared to MgO NPs. Our previous studies [14,18] have also demonstrated that by coating fluorescent emitters with silk (or other polymers with refractive index of the polymer n_pol_ > 1 greater than air), superior emission rates are observed. Our numerical and theoretical models showed that for emitters with particle refractive index (n_MgO_ = 1.73 at λ = 650 nm) greater than the refractive index of the surrounding media, enhanced emission is observed.

The increase in brightness was confirmed with quantum efficiency measurements for MgO-SF spheres, by comparing the fluorescence spectra of the hybrid spheres with that of a reference (nitrogen vacancy (NV) color center in nanodiamonds), using Equation (1) [25].
*QE* = *QE_NV_* × *I_nt_/I_ntNV_* × ((1 − 10^−*A*^*_NV_*)/(1 − 10^−*A*^)) × (*n*^2^/*n_NV_*^2^)(1)

The reported value of 0.7 is used for the *QE* of the reference NV center *QE_NV_* in nanodiamonds [25]. *I_nt_* and *I_ntNV_* are the area under the curve for the PL spectra of MgO-SF spheres and the NV center (in diamond) respectively, when excited with 532 nm with a pump power of 100 µW. *A* and *A_NV_* are the absorption coefficients of MgO-SF spheres and the reference nanodiamonds at 532 nm wavelength. *n* and *n_NV_* are the refractive indices of the solvent (water, *n* = *n_NV_* = 1.33) used to suspend the MgO-SF spheres or the nanodiamonds for absorption measurement. The *QE* of MgO-SF spheres from the Equation is hence calculated to be *QE*_MgO-SF_ = 0.52, higher than that of the MgO NPs of *QE*_MgO_ = 0.45 reported in our earlier work [2].

### 3.3. Fluorescence Properties

#### 3.3.1. Confocal Microscopy with Green Excitation

The fluorescence properties for MgO-SF were explored via a custom confocal microscope. Samples were prepared for confocal microscopy by drop casting 50 µL of NP- aqueous solutions (0.5 mg/mL) on Si substrates followed by air drying at room temperature. A continuous wave 532 nm green laser source was used to illuminate the samples at a low excitation power of 80 µW. Fluorescence coarse scans (200 × 200 µm^2^) of MgO, SF and MgO-SF spheres, are shown in Figure 3a–c. The color bars representing the intensity of counts are shown to the right of each image. Both MgO-SF and MgO NPs showed bright 0.5–1 M counts/s, distinct from the background as shown in Figure 3a,b, respectively. Moreover, MgO-SF sample (Figure 3a) shows bright circular regions of fluorescence with significantly low agglomeration, compared to larger agglomerates in case of MgO NPs on their own (Figure 3b).

Silk only spheres of Figure 3c show a false color scale going up to only 50 k counts/s, which is ≈20 orders of magnitude less than the fluorescence of MgO-SF spheres. The inset of Figure 3a,b shows zoomed confocal maps (5 × 5 µm^2^) for MgO-SF and MgO NPs, respectively. SF only spheres showed very low autofluorescence in the zoomed regions, hardly distinguishable from the background, as shown in the inset of Figure 3c. The emission count rates were investigated as a function of time for 10–12 MgO-SF, MgO and SF particles. The count traces for two representative particles for each of MgO-SF (blue), MgO (red) and SF (magenta) NPs are shown in Figure 4 for comparison.

The MgO-SF NPs showed 2-4× brighter counts compared to the uncoated MgO NPs, as also demonstrated by the representative emission rates of Figure 4. In addition, SF spheres showed a low background autofluorescence of the order of 5 kcts/s, which means that the silk’s autofluorescence does not present a background noise. Thus, coating the MgO NPs with SF not only provides stability against agglomeration but also results in enhanced fluorescence count rates. This implies that MgO-SF spheres are suitable for in vitro bioimaging probe and will be discussed in Section 3.4. However, for in vivo bioimaging applications, the photostable materials that fluoresce in higher wavelength are highly sought-after candidates. Thus, the fluorescence characteristics of these samples that were investigated using 633 nm laser wavelength source will be discussed in the following section.

It is to be noted that the number of MgO NPs inside a SF sphere cannot be quantized for the co-flow method utilized for sphere fabrication. However, for the current method, the suitable concentrations of MgO and SF were mixed uniformly together in aqueous and then injected to the system for sphere fabrication. From the initial concentrations used, the ratio of MgO to SF by weight was 1:20. To be consistent with the fluorescence measurements of Figure 3 and Figure 4, we were careful to select the smallest spheres, within the single emitter resolution, only that increases the probability of containing single MgO NPs.

The initial photobleaching of counts occurs in both MgO and MgO-SF spheres. The exponential decay is different for MgO-SF spheres since there are two components exhibiting photobleaching: SF sphere (shell) and MgO NPs embedded inside the SF sphere.

#### 3.3.2. Confocal Microscopy with Red Excitation

Both MgO NPs and MgO-SF spheres show good optical emission properties when excited with longer excitation wavelength 633 nm, as shown in Figure 5. The results show that a longer excitation wavelength of 633 nm laser source was selected to excite the MgO and MgO-SF spheres with 300 µW power. 

MgO NPs provided a weak fluorescence response than that of MgO-SF spheres at 633 nm excitation. The confocal fluorescence map of silk coated MgO NPs (Figure 5a), depicted bright fluorescent spheres of the order of 1 M compared to MgO NPs showing less fluorescence of the order of 0.5 M (Figure 5b). Additionally, the number of bright MgO-SF spheres was greater than weak MgO agglomerates. To determine the photostability, emission traces for fluorescent particles were recorded as shown in Figure 5c. The MgO-SF spheres showed photostable counts till 180 s as shown by the blue trace compared to MgO only NPs (red trace). Moreover, the emission counts for MgO-SF spheres showed a slower decay of counts up to 30% in 180 s, while those of MgO NPs showed a larger drop of up to 40% in 180 s. As observed in Figure 4, the initial photobleaching trend differs for MgO and MgO-SF spheres in Figure 5.

Hence, while MgO NPs on their own show extremely low fluorescence signal, their coating with SF enables their successful excitation and fluorescence collection in the NIR-1 window. This demonstrates the strong potential of these hybrid spheres for in vivo imaging applications. This experiment also demonstrates that the MgO-SF spheres can be the re-excited to produce fluorescence further down in the NIR region, in contrast to traditionally used organic dyes where reabsorption and re-excitation is an issue. Hence SF coated MgO particles fluoresce with improved optical properties.

### 3.4. Cellular Imaging with Wide-Field Microscopy

The cell viability for MgO and MgO-SF NPs was assessed on HaCaT and U87MG cells as shown in Appendix A. The data reveals that there are no negative effects on the cell viability due to the presence of MgO-SF NPs. After confirming the cell viability for these hybrid MgO-SF NPs, their potential for biological imaging was explored via wide-field microscopy. The MgO and MgO-SF NPs were cultured in HaCaT, U87MG and MCF7 cell lines. The Hoechst 33342 dye which is present in the NucBlueTM Live Ready Probes Reagent (emission centered at ≈481 nm) was added to stain the nuclei.

#### 3.4.1. Imaging of HaCaT Cells

Figure 6a,c shows the fluorescence of the stained nuclei (blue), when excited with ≈390 nm. Figure 6b,d demonstrates the fluorescence from MgO and MgO-SF hybrid spheres, respectively, taken up by HaCaT cells after 24 h of incubation. 

The corresponding fluorescence images were collected by exciting the NPs at 560 nm and collecting the fluorescence in red to NIR (600–800 nm) range. A comparison between Figure 6b,d indicates that MgO was partially taken up by the HaCaT cells after 24 h incubation. Figure 6b shows very weak and continuous fluorescence induced by MgO NPs when excited at 560 nm as shown in inset of the figure. In contrast the MgO-SF spheres show higher uptake as represented by the clearly fluorescent bright circular red regions, as shown in Figure 6d. The arrows in the inset indicate the fluorescent bright red spheres present in the cell membrane. To confirm the localization of MgO NPs and MgO-SF spheres in different cellular compartments, we performed colocalization experiments as described in Section 3.6.

The reason for low uptake could be the agglomeration of MgO NPs on their own, increasing their size and lowering their uptake in the cells. In the absence of any bright NP fluorescence, the image of Figure 6b merely shows weak background autofluorescence from the cell membranes. Moreover, due to the bright fluorescence of MgO-SF NPs and higher contrast, the autofluorescence of the cells was negligible.

Hence silk coating enhanced the uptake of the MgO-SF spheres resulting in high contrast imaging and tracking inside the cells. To investigate the potential of MgO-SF spheres as long-term imaging agents, we incubated them in the HaCaT cells for an additional 24 h. After a total of 48 h incubation, the MgO-SF spheres were still fluorescent inside the cell as shown in the Appendix A. Hence SF coating facilitates the uptake of MgO-SF spheres at prolonged times, which allows their facile permeation inside cells [13,15].

Moreover, for NP-based imaging it is very crucial that the cell autofluorescence is avoided, as the autofluorescence will obstruct the fluorescence from the NPs. The bright field image of Appendix A shows the entire cell morphology. A comparison with fluorescence images of Figure 6 indicates a very low auto fluorescence signal from the cells.

#### 3.4.2. Imaging of Brain Cancer Cells

U87MG brain cancer cells lines were also used to test the capability of MgO NPs and MgO-SF spheres for cell imaging. Figure 7a represents cellular uptake imaging of MgO NPs. Figure 7a,c shows the fluorescence of the stained nuclei around 461 nm (blue), when excited with 390 nm. Figure 7b,d demonstrates the fluorescence from MgO and MgO-SF hybrid spheres, respectively, up taken by the U87MG cells after 24 h of incubation. A comparison of the respective NP-fluorescence images indicates that both MgO and MgO-SF were up taken by the U87MG cells. However brighter fluorescence was observed from MgO-SF spheres.

The cellular uptake of both NPs after 48 h incubation is shown by the NP fluorescence images of Appendix A. The MgO NPs on their own showed weaker fluorescence signal and more agglomeration after 48 h of incubation. However, the MgO-SF spheres in contrast showed bright fluorescence and a relatively more even spatial distribution inside the cells even after extended incubation time of 48 h. Moreover, green fluorescence from MgO-SF spheres is also noticed under blue (485 nm) excitation (Appendix A) because silk fibroin has a fluorescence maximum under blue (485 nm) excitation, while 560 nm excitation causes fluoresces of the MgO NPs (Appendix A) in the hybrid MgO-SF spheres.

#### 3.4.3. Wide-Field Imaging of Breast Cancer Cells

The MCF 7 cells were also tested with the NPs as a second type of cancerous cell line. Two types of cancer cells were chosen because, on many occasions, it has been reported that MgO nanoparticles can be selectively toxic towards cancer cells [26,27]. Figure 8a,c demonstrates the presence of live cells, after the uptake of MgO and MgO-SF NPs, respectively, through the nuclei fluorescence. Like brain cancer cells, both MgO and MgO-SF were taken up by the BCCs, as shown by the red fluorescence images of Figure 8b,d. However, MgO-SF spheres appear to be brighter, as indicated by the saturating fluorescence of Figure 8d. Moreover after 48 h of incubation, the hybrid MgO-SF spheres showed significant fluorescence inside the cells as demonstrated by the bright circular regions of Appendix A, as compared to MgO only NPs.

Thus, comparing the three cell lines, MgO-SF spheres were taken up by the noncancerous cells as well as the cancer cells. In contrast, MgO only NPs only indicated uptake by cancer cells, under the same incubation times and experimental conditions. Thus, the MgO NPs are more susceptible for U87MG and BCCs compared to the HaCaT cells. These findings are confirmed with colocalization experiments results presented in Section 3.6.

### 3.5. Mobility Analysis

After establishing that MgO-SF spheres are internalized in both noncancerous and cancerous cells, we measured the intracellular mobilities of these hybrid spheres in the three cell lines. The tracking of MgO-SF spheres in HaCaT, U87MG and MCF7 cell lines was monitored by recording their two-dimensional trajectories under wide-field microscope. Mobility calculations were performed to compare the dynamics of MgO-SF spheres inside the cells. Using a wide-field microscope, videos and images were recorded to calculate the diffusion coefficient by Equation (2) [24].
(2)$$r^{2}=4\;\frac{k_{B}T}{6\pi \eta \alpha }\;\Delta t=\;4D\Delta t$$
where, *k_B_* is the Boltzmann constant, *T* is temperature (37 °C), *η* is viscosity of the medium, *α* is the average radius of spheres, Δt is the time interval, and D is diffusion coefficient. D values were calculated for number of spheres. Higher D value corresponds to higher diffusion coefficient which reveals greater mobility of spheres in cellular environment.

From the above equation, D values of MgO-SF spheres for HaCaT cells ranged from 0.0075 ± 0.0011 to 0.0825 ± 0.0025 µm^2^/s. In contrast very low diffusion values on average D_mgo_ = (1.20 ± 0.24) × 10^−4^ µm^2^/s were observed with MgO alone inside skin keratinocyte cells from our earlier work [2]. This is in agreement with literature where very low diffusion coefficients <0.01 μm^2^ s^−1^ are expected for NPs with diameters equal or greater than 50 nm in cells [24,28]. Literature also shows that NPs coated with lipid, polymers and surfactant show improved cellular mobility [29,30].

However, the use of MgO NPs without any surface modification in many biological applications is limited. The cleaved edges and rough surfaces result in trapping of NPs in endosomes after cellular uptake. Moreover, the direct surface functionalization of NPs for specific biorecognition is not straightforward, which is an obstacle for the use of NPs in biological applications. These limitations are addressed by using biocompatible surface modification of NPs with SF.

Diffusion constants for U87MG cells were found to exist between a minimum of 0.0275 ± 0.0035 µm^2^/s to a maximum of 0.2425 ± 0.0530 µm^2^/s. Inside breast cancer cells the diffusion rate for MgO-SF varies from 0.006 ± 0.002 to 0.5 ± 0.1 µm^2^/s. In contrast the MgO only NPs showed diffusion constant in the range of 0.026 ± 0.006 to 0.177 ± 0.03 µm^2^/s.

A comparison of the diffusion rates inside normal (HaCaT) and cancer (U87MG) cell lines is presented in Figure 9 which implies that MgO-SF spheres are more mobile inside cancer (U87MG) cells than normal (HaCaT) cell lines.

### 3.6. Colocalization

Nanoparticles enter cells by different endocytic pathways and are transported into endosomes via endosomal-lysosomal pathway [31]. Endosomes are relatively large (around a micron) membrane bound compartments produced by plasma membrane and are usually found in the cytoplasm of cells [32].

To confirm the uptake and internalization of MgO-SF, we performed colocalization experiments. The cells were transfected with CellLight^®^ Early Endosomes-GFP, BacMam 2.0, which causes the endosomes to express GFP (green fluorescent protein). The nuclei were stained with Hoechst 33342 dye. Wide-field imaging was performed to image the control cells as well as the cells cultured with MgO NPs and MgO-SF spheres after 2 and 4 h of incubation.

Figure 10 compares the fluorescence images of endosome stained HaCaT cells without NPs (control) with those cultured with NPs. Figure 10a shows the control cells, without NPs, where only nucleus was stained with DAPI. Figure 10b,c shows the cells incubated with MgO NPs after 2 and 4 h of incubation, respectively. Green fluorescence from the stained endosomes can be seen in both images (b,c), however the red NP fluorescence was not observed at both time intervals. This indicates that MgO NPs were not inside the cells at these times points. Figure 10d,e shows the cells incubated with MgO-SF spheres after 2 and 4 h, respectively. After 2h, few MgO-SF spheres (red fluorescence) were seen near the nuclei. Most of the spheres were located in vesicles as can be clearly seen with the overlapping green and red fluorescence spots corresponding to endosomes and MgO-S, respectively, after 4 h incubation.

A similar experiment was performed with U87MG cell lines as shown in Figure 11a–d. The confocal imaging of control sample of U87MG, without any NPs in Figure 11a clearly shows endosome fluorescence (green). Figure 11b,c shows the red fluorescence from the MgO NPs overlapping with green endosome fluorescence while imaging after 2 and 4 h of NP incubation, respectively. This implies that compared to HaCaT cells, MgO NPs demonstrate a higher uptake by the U87MG cells.

Figure 11d shows MgO-SF spheres inside endosomes after 4 h of incubation in U87MG cells demonstrating the internalization of the hybrid spheres. Hence, compared to MgO NPs on their own, MgO-SF spheres exhibited higher uptake compared to MgO NPs for the HaCaT cell line, due to the uniform spherical size, low agglomeration and SF natural protein coating. However, in the case of U87MG cancer cell line both MgO NPs and MgO-SF spheres showed significant uptake and internalization which is most likely due to the higher susceptibility of the cancer cells to engulf foreign agents [33]. 

Therefore, in addition to reduced agglomeration, increased mobility and brightness, another advantage of MgO-SF spheres over MgO NPs (on their own) is increased cellular uptake. This is consistent with published reports that have shown that other nanoparticles such as nanodiamonds and quantum dots coated with lipid layers internalized into cells easier than non-coated NPs [29,30,33,34]. However, SF holds special significance due to its biodegradability and bio-resorbability in comparison to other coating materials [35]. Although the MgO-SF spheres were taken up by the cells, they did not cause any observable adverse effects as presented in Appendix A. These spheres also maintain photostable fluorescence inside both cell types, important for future bioimaging applications. Similar findings were reported with nanodiamond-SF nanospheres [15]. The full potential of these nanoparticles can only be relieved by their further testing through confocal studies of tissues injected with MgO-SF spheres, followed by in vivo imaging studies.

## 4. Conclusions

The paradigm shift offered by this report is to move from the use of poorly targeted and toxic nanoparticles to autofluorescent, biocompatible and furthermore biodegradable nanoparticles. Whilst green synthesis approaches are gaining the attention of academia, green materials and methods such suggested here are also needed to be promoted into biomedical practice. In this study, MgO NPs were encapsulated with SF protein via microfluidics to produce hybrid spheres. The structural properties revealed a uniform spherical morphology and reduced agglomeration of the MgO-SF spheres compared to the bare MgO NPs. The optical properties with two different excitation wavelengths showed enhanced fluorescence and higher quantum yield for the spheres. The presence of multiple color centers with emission centered in the visible as well as NIR explored the potential of these spheres for both in vitro and in vivo imaging applications. Time dependent cell viability data on human skin and human brain glioma cells showed that the spheres did not introduce any significant level of cytotoxicity. Real time wide-field fluorescence imaging revealed the potential of the hybrid MgO-SF spheres as bright cell imaging tools for HaCaT, U87MG and MCF7 cell lines. Diffusion and colocalization experiments revealed that MgO-SF spheres showed higher uptake and increased intracellular mobility in the cytoplasm of the cells compared to bare MgO NPs. In addition to biocompatibility and biodegradability, the main advantage of these complex optical structures, consisting of the natural biopolymer silk and fluorescent MgO, is possessing dual functionalities, drug delivery and bioimaging. Ultimately, it is expected that these initial results will be the first step towards the widespread use of biodegradable MgO-SF spheres as fluorescent markers for short term bioimaging and controlled drug delivery.

## Figures and Tables

**Figure 1 nanomaterials-11-00695-f001:**
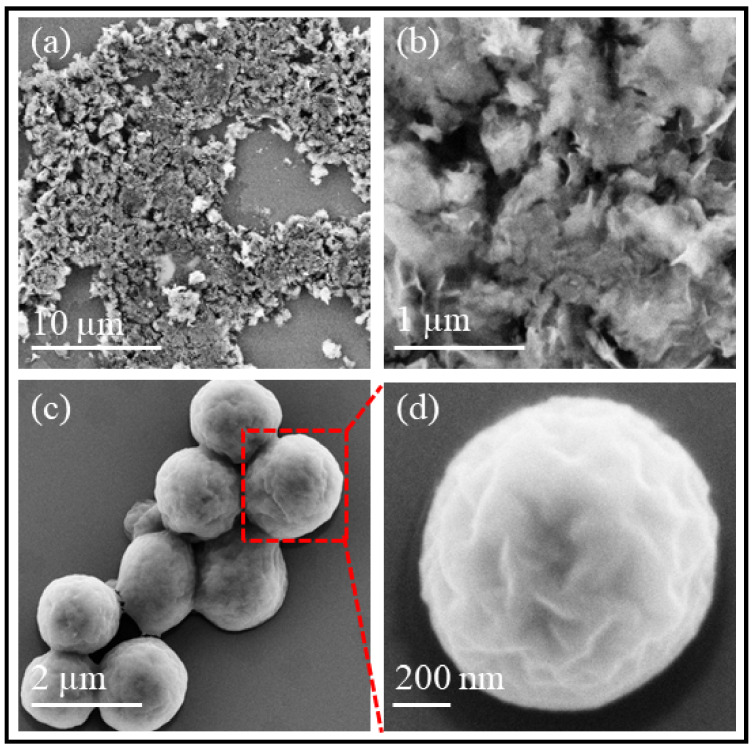
FESEM micrographs of (**a**,**b**) MgO nanoparticles (NPs) and (**c**,**d**) MgO-SF spheres at different magnifications. Magnification of (**a**) is 3×, (**b**) is 30×, **c** is 30× and (**d**) is 100×. Note that the size of the spheres was reduced as mentioned in Table 1.

**Figure 2 nanomaterials-11-00695-f002:**
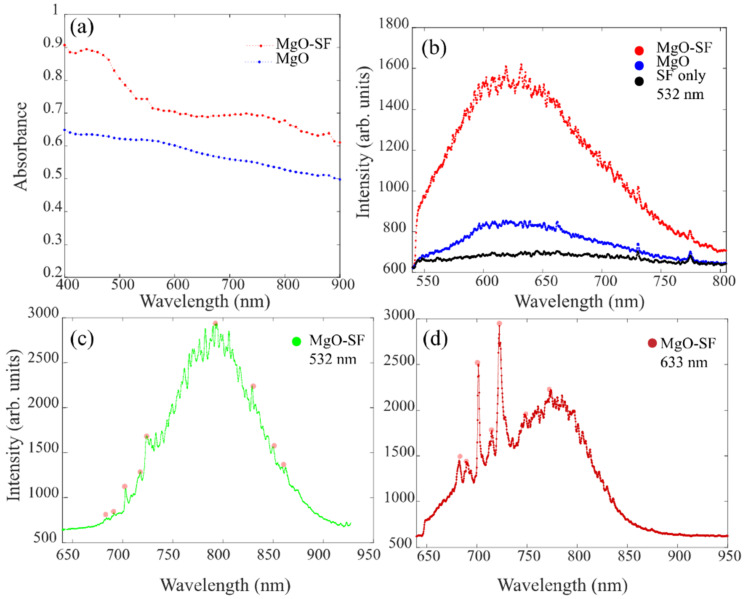
Room temperature (**a**) absorption spectra of MgO-SF and MgO NPs both with a concentration of 1 mg/mL. The absorption measurements are background subtracted. Any scattering errors due to the solution are subtracted before plotting. (**b**) Photoluminescence (PL) spectra of MgO NPs and MgO-SF and SF spheres on silica substrate, excited with 532 nm. PL spectra from a stable less occurring emission center in MgO-SF NPs, excited with (**c**) 532 nm and (**d**) 633 nm wavelengths. There is no difference in experimental methodology. The centers are randomly distributed at different points throughout the samples.

**Figure 3 nanomaterials-11-00695-f003:**
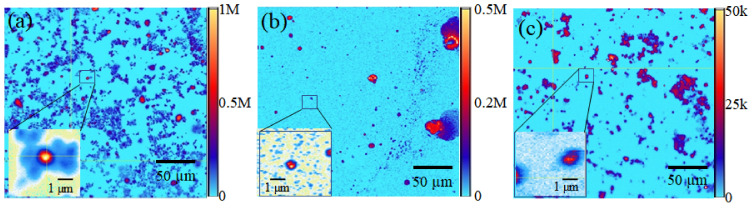
Two-dimensional confocal fluorescence maps (200 × 200 µm^2^) of (**a**) MgO-SF, (**b**) MgO, and (**c**) SF only spheres, excited with 80 µW. Zoomed fluorescence scans (5 × 5 µm^2^) of MgO-SF (max counts up to 1 M), MgO (max counts up to 0.5 M) and SF spheres (low counts up to 50 k) shown in the insets of (**a**–**c**) respectively.

**Figure 4 nanomaterials-11-00695-f004:**
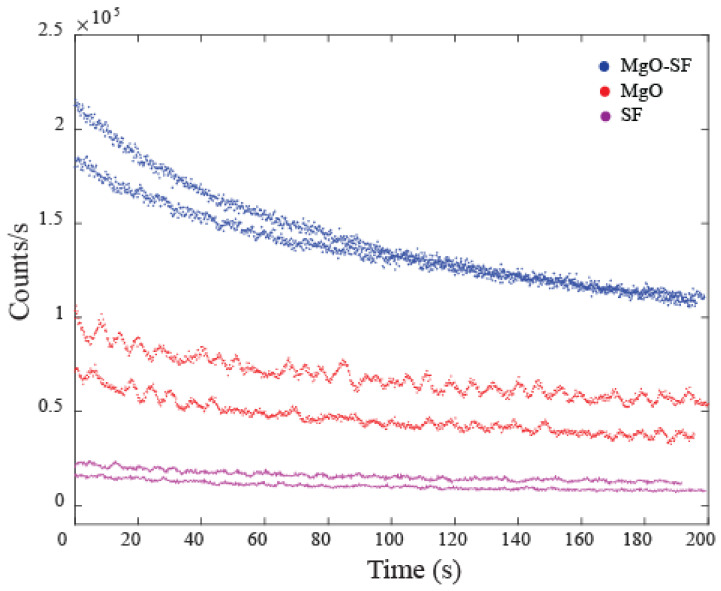
A comparison curve of emission traces of MgO-SF (blue), MgO (red) and SF (magenta) spheres.

**Figure 5 nanomaterials-11-00695-f005:**
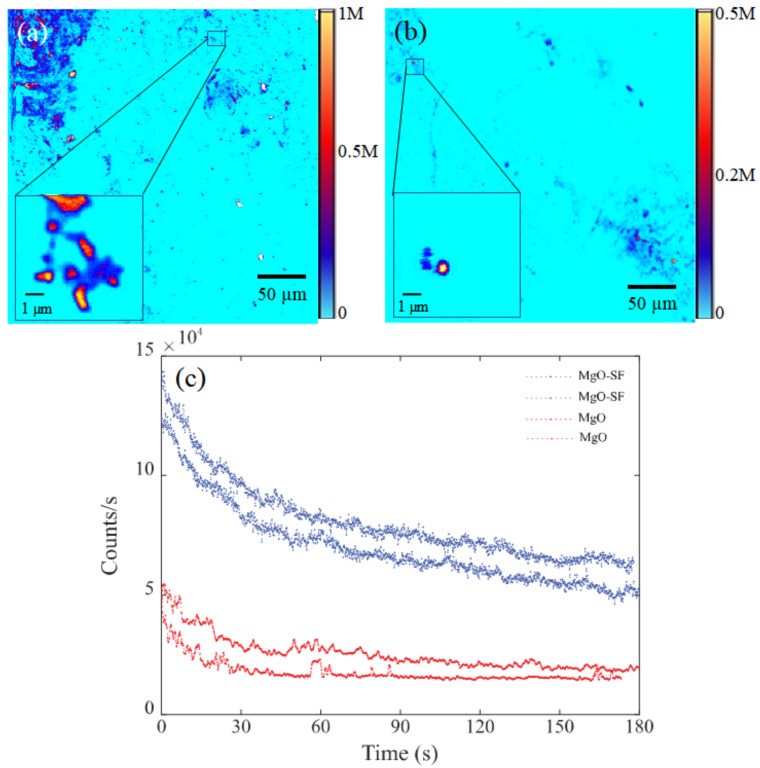
A 200 × 200 µm^2^ (coarse) confocal fluorescence map of (**a**) MgO-SF spheres and (**b**) MgO NPs excited with 633 nm excitation wavelength at 300 µW. The inset shows a zoomed image (10 × 10 µm^2^) for a representative particle highlighted (box). (**c**) Count traces as a function of time for MgO-SF (blue) and MgO NPs (red).

**Figure 6 nanomaterials-11-00695-f006:**
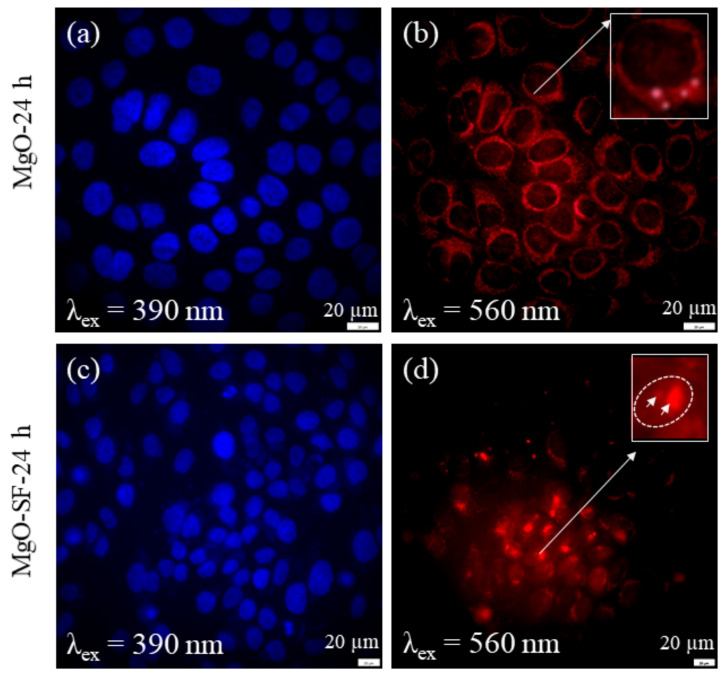
Wide-field fluorescence images of HaCaT cells incubated for 24 h with 5 μg/mL of (**a**,**b**) MgO NPs, and (**c**,**d**) MgO-SF NPs under 390 and 560 nm excitation. No distinct NP fluorescence was seen in MgO cultured cells. (White box highlights the very weak fluorescence from three MgO NPs.) The arrows in inset (**d**) indicate two fluorescent, bright red MgO-SF spheres.

**Figure 7 nanomaterials-11-00695-f007:**
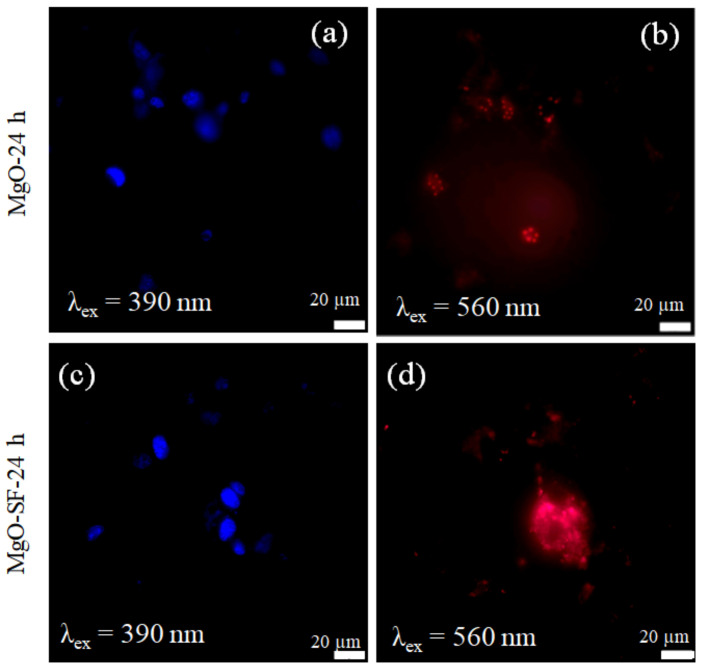
Wide-field images of U87MG cells incubated with 5 μg/mL of (**a**,**b**) MgO NPs, (**c**,**d**) MgO-SF spheres at 37 °C for 24 h. The presence of live cells in indicated by the blue fluorescence of the stained nuclei. The images in **a**,**c** show the fluorescence of the stained nuclei (blue), when excited with 390 nm, while those of **b**,**d** demonstrates the fluorescence from MgO and MgO-SF hybrid spheres.

**Figure 8 nanomaterials-11-00695-f008:**
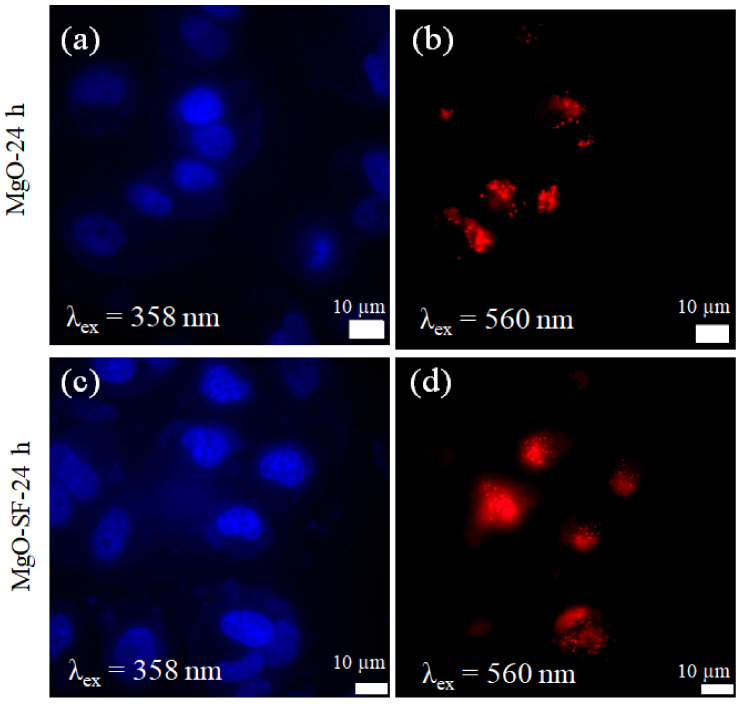
Wide-field images of breast cancer cells incubated with 5 μg/mL of (**a**,**b**) MgO NPs, (**c**,**d**) MgO-SF spheres under 390 and 560 nm excitation and imaged after 24 h.

**Figure 9 nanomaterials-11-00695-f009:**
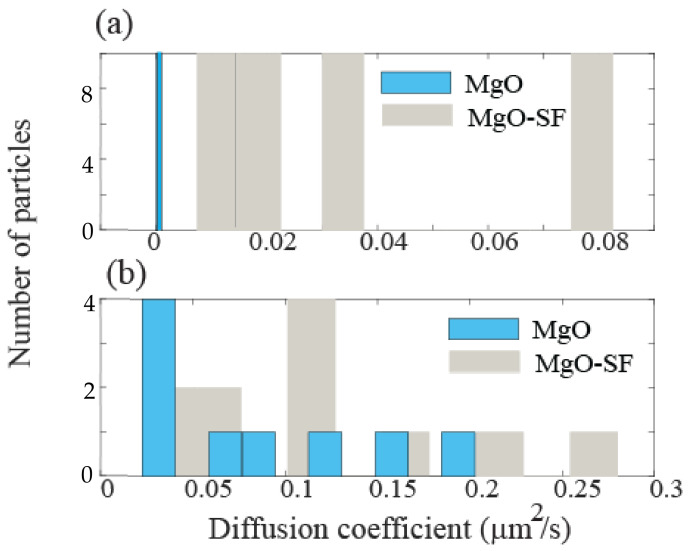
Histogram plots for the comparison of diffusion coefficient of MgO-SF spheres in (**a**) HaCaT and (**b**) U87MG cells.

**Figure 10 nanomaterials-11-00695-f010:**
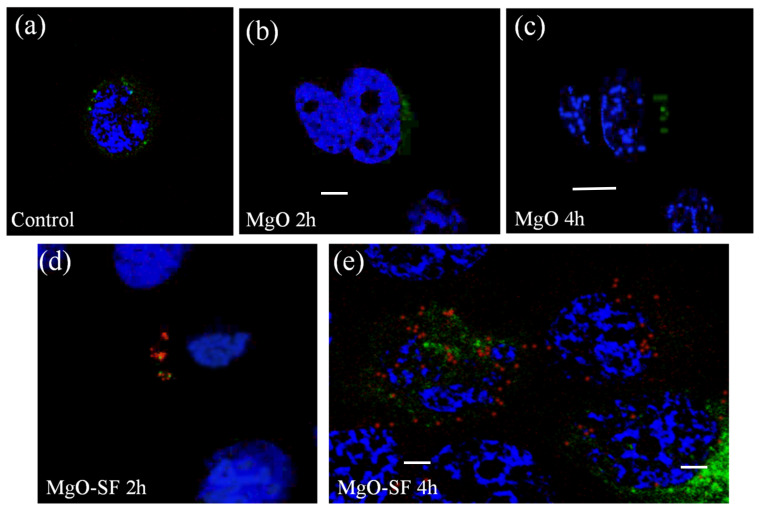
Confocal fluorescence images of endosome stained samples. (**a**) Untreated HaCaT cells (control). HaCaT cells cultured with MgO after (**b**) 2 and (**c**) 4 h of incubation. Cells cultured with MgO-SF after (**d**) 2 and (**e**) 4 h of incubation. Endosome excited at 485 nm wavelength, fluoresce in green, stained nuclei excited at 390 nm fluoresce in blue, while NPs have a distinct red fluorescence when excited at 560 nm. (Repeated excitation of the cells at different time points caused photobleaching of the stained nuclei.) The scale bars are 5 µm each.

**Figure 11 nanomaterials-11-00695-f011:**
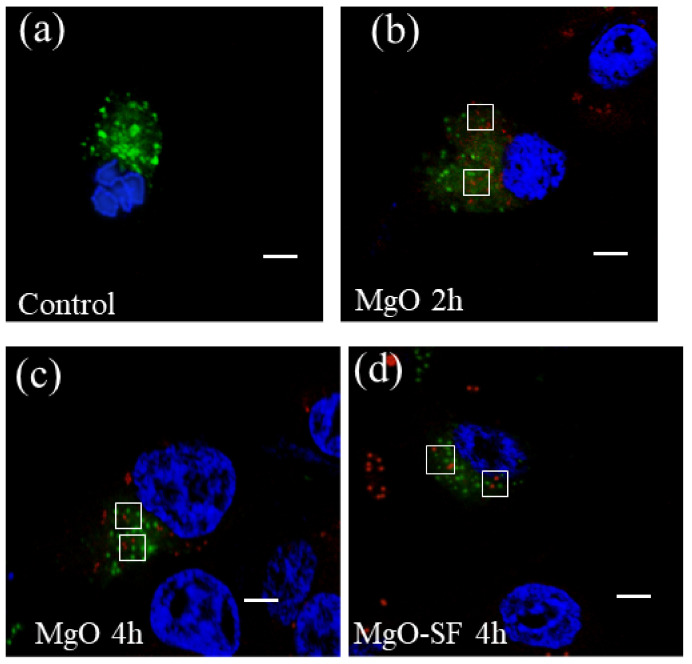
Confocal fluorescence images of endosome stained samples. (**a**) Untreated U87MG cells, (**b**) MgO cultured U87MG cells after (**b**) 2 and (**c**) 4 h of incubation. (**d**) MgO-SF cultured U87MG cells after 4 h of incubation. MgO NPs were internalized by the cells both after short and long incubation time. White squares in (**b**–**d**) represent the region where red and green spots are overlapping. The scale bars are 5 µm each.

**Table 1 nanomaterials-11-00695-t001:** In solution particle size distribution via DLS measurements on MgO and MgO-SF samples.

Sample	Solution	Hydrodynamic Diameter	PDI
MgO-SF	PBS	280 ± 10 nm	0.99 ± 0.01
MgO-SF	PBS+10% FBS	180 ± 7 nm	0.96 ± 0.03
MgO NPs [2]	PBS	1.0 ± 0.5 µm	0.25 ± 0.01
MgO NPs [2]	PBS+10% FBS	600 ± 200 nm	0.95 ± 0.02

## Data Availability

The data is available on reasonable request from the corresponding author.

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
