# Peer review of "Silk Fibroin Coated Magnesium Oxide Nanospheres: A Biocompatible and Biodegradable Tool for Noninvasive Bioimaging Applications"

_nanomaterials, 2021, doi:10.3390/nano11030695_

Round 1

Reviewer 1 Report

The article “Silk fibroin coated magnesium oxide nanospheres: A Biocompatible and biodegradable tool for non-invasive bioimaging applications” by Jitao Li et al. describes the way to prepare fluorescent magnesium oxide nanoparticles coated with silk fibroin and use them as a fluorescent probe for in vitro bioimaging. The importance of this work is in demonstrating a new example of nontoxic, biocompatible and fluorescent nanoparticles with potential for biomedical applications. The behavior of the nanoparticles and their interaction with different cell lines was thoroughly investigated by confocal and wide field microscopy and findings support ideas of the authors. However, there are several flaws in the manuscript, which prevent it to be accepted in the present form.

  1. The authors stated in the text what fluorescence of the nanoparticles comes from “O-vacancy related centres” and from Cr3+ and V2+ ions presented in the magnesium oxide core. Therefore, it is more correctly to describe the material as Cr3+ and V2+ doped MgO. Information about Cr and V content is absent in the Title and the Abstract of the manuscript and should be added.
  2. The presence of the different metal ions in the nanoparticles should be confirmed by a well-established analytical technique such as ICP-MS.
  3. In the Introduction section, lines 53-66, the authors argue that application synthetic polymers for preparation of drug delivery carriers is undesirable. However, for the synthesis of nanospheres they use polyvinyl alcohol, which is synthetic polymer. In addition, it is unclear what exactly the authors meant in fragment of the text on lines 52-61: “However, majority of the polymers have some inherent short comings such as hydrophobicity, or the need to be dissolved in organic solvents with acidic degradation products. The byproducts of these solvents or the polymer itself can cause denaturation and loss of bioactivity of the drugs.” What kind of “byproducts of these solvents” and “acid degradation products” were implied? Probably more details should be added to this part to make it understandable.
  4. 1 Materials section is incomplete. Many reagents used in Section 2 are not fully described, such as polyvinyl alcohol, silk fibroin etc. Characteristics such as molecular weight, purity and supplier should be added.
  5. In section 2.2. ratio of the starting solutions should be added as well as the loading and final concentration of the produced nanoparticles. g force rather than RPMs should be used for description of the synthesis.
  6. In 2.2.1. two different processes are mentioned. “Reduction of size” on the line 107 and “reduce of the size distribution” on the line 108. Which one is correct?
  7. In Structural characteristics section the authors determined size and size distribution of MgO-SF nanospheres from FESEM images but determination of size reduced MgO-SF was done by DLS. Additionally no micrographs of size reduced nanospheres presented in the article. Is where a reason to use two different techniques for size determination? Is it possible to add a micrograph of size reduced nanospheres to the Figure 1?
  8. A few typing errors are presented in the manuscript:
  • Several references in the text are not formatted properly. See line 41 “cytotoxycity5” and the same at lines 46, 70, 97, 207, 208, 244, 249, 280
  • References section should be carefully rechecked and formatted in accordance with Nanomaterials guidelines. See for example line 571 “p 269”. Also some of journal names abbreviated and some not cf. line 581 “Chem Soc Rev” and line 591 “Journal of Photochemistry and Photobiology B: Biology”
  • Line 556 figure S5 mentioned twice.

Author Response

1.1 The authors stated in the text what fluorescence of the nanoparticles comes from “O-vacancy related centres” and from Cr3+ and V2+ ions presented in the magnesium oxide core. Therefore, it is more correctly to describe the material as Cr3+ and V2+ doped MgO. Information about Cr and V content is absent in the Title and the Abstract of the manuscript and should be added.

Response 1.1: The reviewer is correct pointing out that fluorescence from three different optical centres is recorded in our study. As explained in the manuscript, the Cr3+ and V2+ ions present in the magnesium oxide are results of impurities induced in fabrication process and not intentional doping. As suggested by the reviewer these centres are now introduced in our Abstract:

“This work reports the fabrication and characterization of silk fibroin (SF) coated magnesium oxide (MgO) nanospheres, containing oxygen, Cr3+ and Vn2+ related optical defects, as a nontoxic and biodegradable hybrid platform for bioimaging applications.

1.2 The presence of the different metal ions in the nanoparticles should be confirmed by a well-established analytical technique such as ICP-MS.

Response 1.2: We have conducted the inductively coupled plasma mass spectrometry (ICP-MS) on the MgO nanopowder and the results are published in reference [2]: (Khalid, et al., Biocompatible and Biodegradable Magnesium Oxide Nanoparticles with In Vitro Photostable Near-Infrared Emission: Short-Term Fluorescent Markers. Nanomaterials 2019, 9 (10), 1360). Indeed, according to the commercial supplier, arc melting is known to introduce impurities of vanadium and chromium. These results are presented in Table 2 of reference [2], where each possible impurity is recorded as a percentage present within the sample.

As suggested by the reviewer we have added clarification on the origin of these centres on page 8 section 3.2, line 300-302: It has been shown that arc melting is known to introduce impurities of vanadium and chromium [2]. The inductively coupled plasma mass spectrometry (ICP-MS) on the MgO nanopowder confirms these findings [2].

1.3 In the Introduction section, lines 53-66, the authors argue that application synthetic polymers for preparation of drug delivery carriers is undesirable. However, for the synthesis of nanospheres they use polyvinyl alcohol, which is synthetic polymer. In addition, it is unclear what exactly the authors meant in fragment of the text on lines 52-61: “However, majority of the polymers have some inherent short comings such as hydrophobicity, or the need to be dissolved in organic solvents with acidic degradation products. The byproducts of these solvents or the polymer itself can cause denaturation and loss of bioactivity of the drugs.” What kind of “byproducts of these solvents” and “acid degradation products” were implied? Probably more details should be added to this part to make it understandable.

Response 1.3: We appreciate reviewer’s comment and have revised the paragraph (line 75-78) for clarity as below:

In order to enhance the structural and optical properties of fluorescent NPs [12] as well as cellular uptake [13], polymer coatings have been used in several studies [12-14]. Moreover, polymeric particles and shells on their own have been investigated as targeted drug delivery vehicles to overcome the blood brain barrier and deliver the drug more efficiently [15]. Both natural and synthetic materials can be used for NP coating as well as drug delivery vehicles. The limitation with synthetic polymers is that they are non-biodegradable and hydrophobic. They additionally require harsh acidic or organic solvents, which can induce cytotoxicity [16]. Naturally derived degradable polymers, such as collagen, gelatin, cellulose, hyaluronic acid, alginate, and chitosan have good biocompatibility but have uncontrollable degradation rates, show background autofluorescence and are either opaque or partially transparent to visible light [16]. Hence low transmission will block the NP-fluorescence and the polymer’s autofluorescence will introduce a significant background noise during imaging.

1.4 Materials section is incomplete. Many reagents used in Section 2 are not fully described, such as polyvinyl alcohol, silk fibroin etc. Characteristics such as molecular weight, purity and supplier should be added.

Response 1.4: Polyvinyl alcohol (PVA) with a MW of 30,000-70,000 was obtained by Sigma Aldrich. Silk fibroin solution with a weight to volume percentage of 6.5% was extracted from Bombyx Mori silk cocoons as mentioned in the literature [18].

1.5 In section 2.2. ratio of the starting solutions should be added as well as the loading and final concentration of the produced nanoparticles. g force rather than RPMs should be used for description of the synthesis.

Response 1.5: According to the reviewer;s suggestion, we have modified the Section as follows:

2.2. Synthesis of MgO-SF spheres: For the fabrication of MgO-SF spheres a 2 mg/mL solution of MgO NPs was prepared. 1 mL of SF aqueous solution (from 65 mg/mL as prepared concentration) was mixed with 5 mL of MgO aqueous solution (from 2 mg/mL concentration) and 0.5 mL of miliQ water. The prepared mixture hence contained 10 mg of SF and ~1.5 mg of MgO NPs, per mL of the solution, giving a weight percentage of 10:1 for SF versus MgO. MgO-SF spheres were fabricated through a co-flow microfluidic system as reported in our previous work [13]. The working solution of MgO-SF was injected through an inner smaller diameter (30G) needle of a co-flow device, dispensing at a discrete flow rate of 40 μL/h. The polymer polyvinyl alcohol (PVA) with a 50mg/mL concentration was loaded in a syringe with bigger diameter (16G) and dispensed at a continuous flow rate of 4 mL/h. The MgO-SF droplets condense to spheres upon contact with PVA stream and were collected in polyvinyl alcohol (PVA) for 10 hours. The final concentration of collected spheres hence contained 400 µL of MgO-SF spheres, with ~0.6 mg of MgO and 4 mg SF content. The spheres-suspension was dried overnight to allow the water to evaporate further decreasing sphere size. The PVA was completely removed via centrifugation twice in water at 40,000 RCF (g-force) for 20 minutes each at 4°C. The washed spheres were then suspended and stored in 2-4 mL of water or phosphate buffered saline (PBS).

1.6 In 2.2.1. two different processes are mentioned. “Reduction of size” on the line 107 and “reduce of the size distribution” on the line 108. Which one is correct?

Response 1.6: This comment provided us the opportunity to rearrange some of the text in the mentioned section as below:

2.2.1 Size-reduced MgO-SF spheres: The limitation of this methodology is that size is of the order of 0.5-1 µm, which is not suitable for in vitro experiments. To reduce the size distribution, the spheres were suspended in PBS, followed by incubation and constantly shaking at 37 ºC. This resulted in size-reduced spheres due to the degradation mechanism in SF. The degradation of the SF in PBS is attributed to bulk swelling of the SF causing silk molecules to detach from the surface reducing the diameter of the spheres overtime. The spheres were incubated for 48 h to make their size suitable for uptake by the cells.

1.7 In Structural characteristics section the authors determined size and size distribution of MgO-SF nanospheres from FESEM images but determination of size reduced MgO-SF was done by DLS. Additionally no micrographs of size reduced nanospheres presented in the article. Is where a reason to use two different techniques for size determination? Is it possible to add a micrograph of size reduced nanospheres to the Figure 1?

Response 1.7: We have added the micrograph of size-reduced spheres to the SI as suggested by the reviewer. The following is added to Section 3,1, line 246-247.

The SEM image of the size-reduced spheres (dried on Si substrate) showed a size distribution of 200±50 nm (Fig S1 (c) in SI ).

(Please see the picture in word file attached)

Figure S1: (c) FESEM micrograph of size-reduced spheres at 40× magnification.

1.8 A few typing errors are presented in the manuscript:

  • Several references in the text are not formatted properly. See line 41 “cytotoxycity5” and the same at lines 46, 70, 97, 207, 208, 244, 249, 280
  • References section should be carefully rechecked and formatted in accordance with Nanomaterials guidelines. See for example line 571 “p 269”. Also some of journal names abbreviated and some not cf. line 581 “Chem Soc Rev” and line 591 “Journal of Photochemistry and Photobiology B: Biology”
  • Line 556 figure S5 mentioned twice.

Response 1.8: As suggested by the reviewer, we have checked all references and fixed the Fig S5 repetition.

Reviewer 2 Report

The authors present in this paper the characterization of magnesium iodide nanospheres coated with silk fibroin, in order to obtain nontoxic, biodegradable nanotools for non-invasive bioimaging applications.

Though potentially very interesting and useful, I have some concerns about the work presented in the manuscript that I think should be addressed before considering the paper for publication.

  1. What is the radial resolution of the microscope used? This information is crucial in order to establish whether the spot the authors observe in their images do actually represent single molecules or not.
  2. I’m no expert in the field, but could the authors explain why they observe a reduction of the hydrodynamic radius in 10%FBS and to what do they ascribe the smallest peak below 10nm?
  3. I have some troubles fully understanding the results of their absorption and fluorescence experiments:

Concerning the absorption spectra, those shown in Figure 2(a) do not exhibit any peak. It appears that there might have been present some interesting features in the MgO-SF spectrum below 400nm, therefore I would suggest extending the acquisition at least to 250/200nm. On the other hand, the MgO spectrum, in my opinion and according to my experience, does not show any absorption peak but rather a behavior that might be due to scattering of the solution, since what is actually measured in this kind of experiments is the extinction cross-section, given by the combination of the absorption and diffusion processes. Have the authors tried to fit the data to a l-4 power law, typical of the scattering process? That would allow to subtract the diffusion contribution to the spectrum and enlighten and evaluate the eventual existence of absorption peaks.

Concerning instead the fluorescence measurements, could the author please explain more in detail how did they achieve the spectra form the less occurring emission center? I mean, what is the difference, from an experimental point of view, that allows to obtain the spectra in Figure 2(b) or in Figure 2(c)?

Could the authors explain why they observe such an enhancement in the fluorescence emission when silk fibroin is added to MgO nanospheres?

  1. As for the confocal microscopy experiments, how can the authors be sure that no aggregates are formed when air drying the samples? The authors comment on the signal they measure, but the counts they read can be compared among the samples only if they all belong to single nanoparticles, otherwise their considerations are meaningless. To come back to my first point, if the resolution of their microscope allows it, they could for example measure the dimensions of the spots they analyze. It is not enough to try and select the smallest spheres they could find in the images (lines 340-341). Concerning the count traces in Figure 4, why do they decrease versus time? Is it a bleaching effect? If that is the case, why is the exponential decay different for different samples? The authors state that they cannon quantify the number of MgO NPs inside a SF sphere, so I don’t understand the point in comparing MgO-SF and MgO nanoparticles.
  2. Similar considerations can be extended to the results reported in Section 3.3.2.
  3. At line 363 the authors state that “This demonstrates the strong potential of these hybrid spheres for in vivo imaging applications. This experiment also demonstrates that the MgO-SF spheres can be then re-excited to produce fluorescence further down in the NIR region…”, though it is not clear to this reader how the presented results can lead to their second claim; could they please explain the point more in detail?
  4. What is the origin of the autofluorescence observed in Figure 6? Where is collected the fluorescence under the different excitation? Are all the images presented following the same color code (same LUT values)? In the red images in Figure 6, MgO and MgO-SF nanoparticles appears to localize in different cellular compartments, could the authors comment on this point?
  5. Figure S6(d) does not exist in the supplementary material, and the text should be modified accordingly. Moreover, the authors should show the green SF emission under blue excitation among the fluorescence spectra in Figure 2(b).
  6. How many time frames were used to measure the value of the diffusion coefficient? What is the temporal resolution of the Hamamatsu camera they used? To better recognize the possible existence of a particular distribution, I think the authors should analyze a larger number of particles, since at the moment their claim on lines 481-83 is hardly supported by their data.
  7. In the description of the colocalization results, they should verify their description of Figure 10, since they appear to have inverted (d) and (c).
  8. Finally, I’m not sure about the localization they claim to see in Figure 11, since the red and green spots are well separated in the pictures, could the authors clarify?

Minor points to be checked:

  • On line 259 the authors state that the peaks above 800nm are known to correspond to the V2+ ions, and I think they should add a reference to support this statement.
  • The caption of Figure 2(a) reports “…excited with 285 nm”, and since it refers to absorption measurements, I think it should be corrected.
  • The counts reported in Figure 3 are different from those shown in the images.
  • The authors should also add the wavelength range where they collected the fluorescence signal.
  • In all the microscopy images the authors should report the dimensions of the images and of the eventual insets (field of view, number of pixels...).
  • The excitation wavelengths reported in Figure S5, Figure S7 and Figure 8 are different in the caption and in the picture, the authors should verify which is the correct one (390 nm I think).
  • The brain images the authors have shown are not of the best quality, they could provide better pictures, with more cells.
  • In Figure 9, I don’t think the histogram in the top panel should show a fractional number of particles, and I’m not sure if the different width of the bars has any significance.
  • In Figure 10 and 11 there are no dimension bars inside the pictures.

Finally, a thorough revision of the manuscript style and grammar is strongly recommended.

Author Response

Reviewer 2

2.1 What is the radial resolution of the microscope used? This information is crucial in order to establish whether the spot the authors observe in their images do actually represent single molecules or not.

Response 2.1: According to the suggestion, we have added the highlighted text to Methods section 2.5 in the revised manuscript:

The spatial resolution of the confocal microscope used is 0.3 µm. With the diffraction limited customized confocal microscope, nanoparticles with size range down to 10 nm can be scanned through their fluorescence.

2.2 could the authors explain why they observe a reduction of the hydrodynamic radius in 10%FBS and to what do they ascribe the smallest peak below 10nm?

Response 2.2: The reduction in the radius of spheres upon incubation is due to the degradation of silk. The degradation is accelerated by the phosphate ions in PBS because the positive and negative charges more effectively interfere with the β-sheet structure of the silk molecules causing them to swell at different rates compared to water [13].  

The above highlighted text is added to the Section 3.1, line 240-244.

The smaller peak is due to the scattering from the FBS protein aggregates. We have clarified this in SI, Figure S1, caption.

(Please see picture in the attached word response).

2.3 I have some troubles fully understanding the results of their absorption and fluorescence experiments:

(a) Concerning the absorption spectra, those shown in Figure 2(a) do not exhibit any peak. It appears that there might have been present some interesting features in the MgO-SF spectrum below 400nm, therefore I would suggest extending the acquisition at least to 250/200nm.

(b) On the other hand, the MgO spectrum, in my opinion and according to my experience, does not show any absorption peak but rather a behavior that might be due to scattering of the solution, since what is actually measured in this kind of experiments is the extinction cross-section, given by the combination of the absorption and diffusion processes. Have the authors tried to fit the data to a l-4 power law, typical of the scattering process? That would allow to subtract the diffusion contribution to the spectrum and enlighten and evaluate the eventual existence of absorption peaks.

(c) Concerning instead the fluorescence measurements, could the author please explain more in detail how did they achieve the spectra form the less occurring emission center? I mean, what is the difference, from an experimental point of view, that allows to obtain the spectra in Figure 2(b) or in Figure 2(c)?

(d) Could the authors explain why they observe such an enhancement in the fluorescence emission when silk fibroin is added to MgO nanospheres?

Response 2.3 (a): Even though the features in UV region are of interest for material science, it is crucial for bio applications for a fluorescent marker to absorb light at wavelengths longer than 500 nm and to emit light at wavelengths longer than 600 nm to avoid cell’s autofluorescence[1]. In order to emphasize this fact, we added in our introduction:

It is crucial for in vitro experiments for a fluorescent marker to absorb light at wavelengths longer than 500 nm and to emit light at wavelengths longer than 600 nm to avoid cell’s autofluorescence [new ref 6]

Response 2.3 (b): The absorption measurements are background subtracted. Any scattering errors due to the solution are subtracted before plotting.

The text is added to the Figure 1(a) caption.

Response 2.3 (c) There is no difference in experimental methodology. The centres are randomly distributed at different points throughout the samples.

The text is added to the caption of Figure 2, Section 3.2.

Response 2.3 (d): Our previous studies have demonstrated that by coating fluorescent emitters with silk (or other polymers with refractive index of the polymer npol>1 greater than air), superior emission rates are observed. Our numerical and theoretical models have shown that for emitters with particle refractive index (nMgO=1.73 at 650 nm) greater than the refractive index of the surrounding media, enhanced emission is observed.

The above text is added to the Section 3.2 Spectral studies, line 322-327.

2.4. (a) As for the confocal microscopy experiments, how can the authors be sure that no aggregates are formed when air drying the samples? The authors comment on the signal they measure, but the counts they read can be compared among the samples only if they all belong to single nanoparticles, otherwise their considerations are meaningless. (b) To come back to my first point, if the resolution of their microscope allows it, they could for example measure the dimensions of the spots they analyze. It is not enough to try and select the smallest spheres they could find in the images (lines 340-341). (b) Concerning the count traces in Figure 4, why do they decrease versus time? Is it a bleaching effect? If that is the case, why is the exponential decay different for different samples? (c) The authors state that they cannon quantify the number of MgO NPs inside a SF sphere, so I don’t understand the point in comparing MgO-SF and MgO nanoparticles. Similar considerations can be extended to the results reported in Section 3.3.2.

Response 2.4 (a),(c): The confocal microscope we used for the experiment has a gaussian fitting function that maximizes the brightness of the single emitters/NPs with respect to their position. The gaussian fitting returns position co-ordinates x≤0.3 µm, y x≤0.3 µm for individual NPs. The x,y co-ordinates greater than 0.5  µm indicate aggregate of NPs or multiple fluorescent emitters inside the selected emitting spot.  Hence the resolution of the microscope and the scanning software allow an indirect measurement for the dimensions of the spots as well as the number of NPs we analysed within an emitting spot.

The above text is added to Methods Section 2.5, as highlighted with blue fonts in the revised manuscript.

Response 2.4 (b): The initial photobleaching of counts occurs in both MgO and MgO-SF spheres in Figure 4 and 5. The exponential decay is different for MgO-SF spheres since there are two components exhibiting photobleaching: SF sphere (shell) and MgO NPs embedded inside the SF sphere.

The above description is added to the last paragraph of the revised manuscript Section 3.3.1 (392-397) as well as the second last paragraph of Section 3.3.2 (417-418) and highlighted with blue fonts.

2.5. At line 363 the authors state that “This demonstrates the strong potential of these hybrid spheres for in vivo imaging applications. This experiment also demonstrates that the MgO-SF spheres can be then re-excited to produce fluorescence further down in the NIR region…”, though it is not clear to this reader how the presented results can lead to their second claim; could they please explain the point more in detail?

Response 2.5: As suggested by the reviewer we added in our introduction: For in vivo experiments, the light emission should be in near-infrared region (NIR), 700-900 nm, as it penetrates centimetres into tissue, whereas visible light can only travel microns [ref 7 in manuscript].

2.6. What is the origin of the autofluorescence observed in Figure 6? Where is collected the fluorescence under the different excitation? Are all the images presented following the same color code (same LUT values)? In the red images in Figure 6, MgO and MgO-SF nanoparticles appears to localize in different cellular compartments, could the authors comment on this point?

Response 2.6: Figure 6 shows very weak and continuous fluorescence induced by MgO NPs when excited at 560 nm as shown in inset of Figure 6(b). In contrast the MgO-SF spheres show distinctive bright fluorescence spots. For clarity, an inset has been inserted in Figure 6(d) in the revised manuscript. The arrows in the inset indicate the fluorescent bright red spheres present in the cell membrane. To confirm the localization of MgO NPs and MgO-SF spheres in different cellular compartments, we performed colocalization experiments as described in Section 3.6. The reviewer is correct in assuming that all images presented are following the same color code (same LUT values).

Figure 6. Wide field fluorescence images of HaCaT cells incubated for 24 h with 5 μg/ml of (a,b) MgO NPs, and (c,d) MgO-SF NPs under 390 and 560 nm excitation. No distinct NP fluorescent was seen in MgO cultured cells. The white box of inset (b) shows very weak fluorescence from three MgO NPs and arrows in inset (d) indicate two fluorescent bright red MgO-SF spheres.

The highlighted text and modified figure 6 is added to the Section 3.4 of the revised manuscript.

2.7. Figure S6(d) does not exist in the supplementary material, and the text should be modified accordingly. Moreover, the authors should show the green SF emission under blue excitation among the fluorescence spectra in Figure 2(b).

Response 2.7: In the revised version, the wrong label (Figure S6(d) has been removed from main manuscript (and SI) and the text has been modified in the revised manuscript, as follows on line 477-484:

The cellular uptake of both NPs after 48 h incubation is shown by the NP fluorescence images of Fig S6 (a,b) in SI. The MgO NPs on their own showed weaker fluorescence signal and more agglomeration after 48 h of incubation. However, the MgO-SF spheres in contrast showed bright fluorescence and a relatively more even spatial distribution inside the cells even after extended incubation time of 48 h. Moreover, green fluorescence from MgO-SF spheres is also noticed under blue (485 nm) excitation (Fig S6(c)) because silk fibroin has a fluorescence maximum under blue (485 nm) excitation, while 560 nm excitation causes fluoresces of the MgO NPs (Fig S6(b) in the hybrid MgO-SF spheres.

Please note that the Fig 2(b) indicates fluorescence spectra for MgO, MgO-SF and SF only NPs, all under green excitation.

2.8. How many time frames were used to measure the value of the diffusion coefficient? What is the temporal resolution of the Hamamatsu camera they used? To better recognize the possible existence of a particular distribution, I think the authors should analyze a larger number of particles, since at the moment their claim on lines 481-83 is hardly supported by their data.

Response 2.8: We have revised the Methods section 2.8 and have added the highlighted text below to clarify:

To measure the diffusion coefficient, we recorded movies with the inbuilt camera of Olympus microscope under a magnification of 40×. The movies were uploaded in MATLAB. Each movie comprises of 200-300 frames taken for 60-90s duration. A distribution of 8-12 particles in each frame were traced for the movie duration. The x0,y0 coordinates of the particles in the first frame were taken as origin. The displacement ri = xi – x0, … xn – x0,  of each particle in the successive frame was recorded with respect to its origin. Finally, the displacement squared ri2 was plotted as a function of time for each particle. The slope of the curve provided the diffusion coefficient or mobility of the particles inside the cells.

2.9. In the description of the colocalization results, they should verify their description of Figure 10, since they appear to have inverted (d) and (c).

Response 2.9: In the revised manuscript, the description of colocalization results in Section 3.6 have been modified (line 557-563), as highlighted below:

Fig 10 compares the fluorescence images of endosome stained HaCaT cells without NPs (control) with those cultured with NPs. Fig 10(a) shows the control cells, without NPs, where only nucleus was stained with DAPI. Fig 10 (b,c) show the cells incubated with MgO NPs after 2h and 4 h of incubation respectively. Green fluorescence from the stained endosomes can be seen in both images (b,c), however the red NP fluorescence were not observed at both time intervals. This indicates that MgO NPs were not inside the cells at these times points. Fig 10 (d,e) show the cells incubated with MgO-SF spheres after 2h and 4h respectively.

2.10. Finally, I’m not sure about the localization they claim to see in Figure 11, since the red and green spots are well separated in the pictures, could the authors clarify?

Response 2.10:  

We appreciate reviewer’s comment and for clarification, we have highlighted the regions where red and green spots are overlapping (not spatially separated) in Figure 11 (b-d).

(Please see attached word response for picture)

Figure 11. Confocal fluorescence images of endosome stained samples. (a) untreated U87MG cells, (b) MgO cultured U87MG cells after (b) 2h and (c) 4h of incubation. (d) MgO-SF cultured U87MG cells after 4h of incubation. MgO NPs were internalized by the cells both after short and long incubation time. White squares in (b-d) represents the region where red and green spots are overlapping and not well separated. The scale bars are 5µm each.

2.11 Minor points to be checked:

On line 259 the authors state that the peaks above 800nm are known to correspond to the V2+ ions, and I think they should add a reference to support this statement. (Added)

  • The caption of Figure 2(a) reports “…excited with 285 nm”, and since it refers to absorption measurements, I think it should be corrected. (Deleted from caption of Fig 2a)
  • The counts reported in Figure 3 are different from those shown in the images. (Fixed as highlighted in the caption of Fig 3)
  • The authors should also add the wavelength range where they collected the fluorescence signal. The range is defined by the cut-off filter (mentioned in line 153-155) . We used a notch 532 filter to block the excitation beam to interact from the fluorescence. All fluorescence above 532 nm was collected.
  • In all the microscopy images the authors should report the dimensions of the images and of the eventual insets (field of view, number of pixels...). Coarse images are 200×200 µm2, where as zoomed images are 10×10 µm2, as mentioned in each image. Number of pixels are 2048×2048.
  • The excitation wavelengths reported in Figure S5, Figure S7 and Figure 8 are different in the caption and in the picture, the authors should verify which is the correct one (390 nm I think). Fixed
  • The brain images the authors have shown are not of the best quality, they could provide better pictures, with more cells. (Figure resolution improved)
  • In Figure 9, I don’t think the histogram in the top panel should show a fractional number of particles, and I’m not sure if the different width of the bars has any significance. Fractional number of particles is now removed, as suggested. The width presents the range of values for the x-axis.
  • In Figure 10 and 11 there are no dimension bars inside the pictures. Scale bars are added to Figure 10 and 11 as suggested.

Reviewer 3 Report

Research paper titled ‘Silk fibroin coated magnesium oxide nanospheres: A Biocompatible and biodegradable tool for non-invasive bioimaging applications’ by Li et al submitted to Nanomaterials journal. Authors report synthesis of silk fibroin modified MgO NPs and propose their application for bioimaging. Authors also demonstrated the bioimaging of various cancer cells using the modified MgO NPs. The study is thorough and complete, however, novelty and significance needs to be highlighted (as compared to published studies). I suggest performing major revision.

  • Authors to please indicate innovation and significance of the study in abstract.
  • Some errors in Intro para 1: tunable optical properties4 and low toxicity. Delete ‘4’. Similar error repeated in the last line. Seems like it is the reference citation? It is repeated at several places throughout the manuscript. Please correct.
  • Para 1. Please mention what metal oxides are being used currently? Which ones have limitations and what?
  • MgO has been previously applied for bioimaging. Then what is the novelty and innovation of the current study. Please comment
  • Please comment and include in the manuscript: how is the current paper different, significant (and not incremental) advance and addresses a research gap, as compared to their previous publication : Khalid et al. Nanomaterials 2019, 9 (10), 1360.
  • Has SF stabilized MgO NPs not been reported before?
  • What is meant by ‘dual biodegradable’?
  • MgO NPs were purchased and not synthesized in lab? Please comment. That means only the Silk modification was performed in lab?
  • Please explain the size reduction of spheres in results and discussion.
  • “Culture media by maintaining the electrostatic repulsion between the spheres” Does that mean these spheres will not aggolomerate once in culture media?
  • PL spectra of SF only is shown, it was expected to have no excitation.
  • Was the impact of modified MgO checked on healthy cells? Like osteoblasts or fibroblasts, to check cytotoxicity?
  • Why different types of cancer cells were studied and not just one?
  • The discussion of all data relating to Figure 2-4, can be simplified, ensuring smooth flow of story, even to a non-specific reader.
  • Authors to include some sentences prior to Conclusion on limitations of the study and what future work is needed for further advancements in this field.
  • Conclusion needs improvements – please suggest applications and novelty.

Author Response

Reviewer 3

3.1 Authors to please indicate innovation and significance of the study in abstract.

Response 3.1: The reviewer is correct that we are not indicating innovation and significance of the study enough. Therefore, we have changed our Introduction and Conclusions as explained below. However, it is hard to change our Abstract significantly due to the word limitations. At the end of Abstract, it is added:

This study may also open new avenues to optimize drug delivery through biodegradable silk and provide non-invasive functional imaging feedback on the therapeutic processes through fluorescent MgO.

3.2 Some errors in Intro para 1: tunable optical properties4 and low toxicity. Delete ‘4’. Similar error repeated in the last line. Seems like it is the reference citation? It is repeated at several places throughout the manuscript. Please correct.

Response 3.2: Corrected

3.3 Para 1. Please mention what metal oxides are being used currently? Which ones have limitations and what?

Response 3.3: There is a paragraph on metal oxides usage for bioimaging applications and their limitations in our Introduction:

“Metal oxide NPs [1-3] have been gaining interest as contrast agent for several bioimaging applications owing to their room temperature single photon emission [1, 3], tunable optical properties4 and low toxicity. However, there are several limitations to the use of these metal oxide NPs, as some of them demonstrate low quantum efficiency and brightness [2-3], high agglomeration in cell culture media [2], while others exhibit dose dependent cytotoxicity [5].”

As suggested by the reviewer we added more specification on some of metal oxides in the Introduction as follows:

 One example is zinc oxide that exhibits broadband emission in the visible region, attributed to defects within the crystallographic structure, suitable for in vitro experiments. However, ZnO NPs generally suffer from pure photostability in cellular environment, in form of photobleaching or photo-blinking [1,3].

Even though it would be good to present a complete coverage of all the metal oxide nanomaterials used so far in bioimaging it would by far exceed the frame of this paper but rather fill a book.

3.4 MgO has been previously applied for bioimaging. Then what is the novelty and innovation of the current study. Please comment

Response 3.4: MgO nanoparticles have been used in biomedical studies but mainly because of their antibacterial and anticancer properties. Rarely, MgO NPs have been used for bioimaging.

In our recently published work [2], we have reported intrinsic photostable fluorescence properties of MgO NPs, used for in vitro experiments. However, the main obstacle for future higher-level biomedical applications of MgO NPs are agglomeration and need for more effective cell internalization. Furthermore, as mentioned above, crucial for in vivo application is that the light emission should be in near-infrared region. Even though, MgO NPs have emitting optical centres in this wavelength window, they are only becoming bright enough with silk coating.

Therefore, in order to emphasize those points we added in Introduction Line 64-70:

However, the main obstacle for future higher-level biomedical applications of MgO NPs are agglomeration and need for more effective cell internalization.

Currently there is no fluorescent nanoparticle that satisfies all necessary criteria for short-term in vivo imaging: biocompatibility, biodegradability, photostability, suitable wavelengths of absorbance, fluorescence that differ from tissue autofluorescence, high internalization and low level of agglomeration. In this study we test fluorescent MgO-SF nanoparticles that satisfy all the criteria.

3.5 Please comment and include in the manuscript: how is the current paper different, significant (and not incremental) advance and addresses a research gap, as compared to their previous publication : Khalid et al. Nanomaterials 2019, 9 (10), 1360.

Response 3.5: In addition to previous response we point out, it is expected that these initial results will be the first step towards the widespread use of biodegradable nanoparticles as fluorescent markers for short term bioimaging and drug delivery.

We are grateful for these comments as we have not pointed out the importance of our findings.

Therefore, those points are added in our Conclusions:

The paradigm shift offered by this study is to move from the use of poorly-targeted and toxic nanoparticles to auto-fluorescent, biocompatible and furthermore biodegradable nanoparticles. Whilst green synthesis approaches are gaining the attention of academia, green materials and methods such suggested here are also needed to be promoted into biomedical practice.

3.6 Has SF stabilized MgO NPs not been reported before?

Response 3.6: The literature reveals there hasn’t been a prior study in this area.

3.7 What is meant by ‘dual biodegradable’?

Response 3.7: it means both MgO and SF are biodegradable. The term is clarified on line 102 as highlighted: This opens the way for several future applications of hybrid dual biodegradable MgO-SF system (both MgO and SF are biodegradable).

3.8 MgO NPs were purchased and not synthesized in lab? Please comment. That means only the Silk modification was performed in lab?

Response 3.8: MgO NPs were purchased and their surface was modified in the lab with SF, as mentioned in Section 2.2 (Methods).

3.9 Please explain the size reduction of spheres in results and discussion.

Response 3.9: This comment is already addressed above in response to Reviewer 2’s comment 2.2. The reason for size reduction is added to Section 3.1, line 240-244.

3.10 “Culture media by maintaining the electrostatic repulsion between the spheres” Does that mean these spheres will not aggolomerate once in culture media?

Response 3.10: Spheres show very low agglomeration in culture media, as evident from Fig S1(b). The FBS protein in the culture medium surrounds the spheres providing a lower agglomeration due to electrostatic repulsion.

3.11 PL spectra of SF only is shown, it was expected to have no excitation.

Response 3.11: SF only spheres were expected to show a weak fluorescence as evident from the Fig 2b (purple spectrum) and Fig 4 (black trace).

3.12 Was the impact of modified MgO checked on healthy cells? Like osteoblasts or fibroblasts, to check cytotoxicity?

Response 3.12: In the SI Figure S4, we have presented cytotoxicity test results for both healthy skin keratinocyte (HaCaT) and cancer (U87MG) cells.

3.13 Why different types of cancer cells were studied and not just one?

Response 3.13: The usage of different type of cancer cells was intentional. Explanation is added at the beginning of Section 3.4: Wide-field imaging of Breast cancer cells:

Two types of cancer cells were chosen because, on many occasions it has been reported that MgO nanoparticles can be selectively toxic towards cancer cells [new reference: 35]

3.14 The discussion of all data relating to Figure 2-4, can be simplified, ensuring smooth flow of story, even to a non-specific reader.

Response 3.14: As suggested by the reviewer we have made significant changes (blue highlights) to present the results with clarity.

3.15 Authors to include some sentences prior to Conclusion on limitations of the study and what future work is needed for further advancements in this field.

Response 3.15: As suggested by the reviewer we have added to conclusions:

The full potential of these nanoparticles will be relieved by their further testing through confocal studies of tissues injected with MgO-silk nanoparticles, followed by in vivo imaging studies.

3.16 Conclusion needs improvements – please suggest applications and novelty.

Response 3.16: We have tried to improve Conclusions following reviewer’s suggestions. This paragraph is added to conclusions:

And at the end:

In addition to biocompatibility and biodegradability, the main advantage of these complex optical structures, consisting of the natural biopolymer silk and fluorescent MgO, is possessing dual functionalities, drug delivery and bioimaging. Ultimately, it is expected that these initial results will be the first step towards the widespread use of biodegradable MgO-SF spheres as fluorescent markers for short term bioimaging and controlled drug delivery.

Round 2

Reviewer 1 Report

Dear colleagues,

All issues raised by the reviewer are corrected in tne new version of manuscript. Therefore the manuscript can be accepted in present form.

Reviewer 2 Report

The authors have fully addressed all the points I raised in my first revision, so I think the paper is now suitable for publication in its present form.

Reviewer 3 Report

Authors have successfully closed all questions. I would like to congratulate the authors. The paper should be accepted for publication.